# Satellite-Derived Barrier Response and Recovery Following Natural and Anthropogenic Perturbations, Northern Chandeleur Islands, Louisiana

**Julie C. Bernier *** [ID]**, Jennifer L. Miselis and Nathaniel G. Plant**

U.S. Geological Survey, St. Petersburg Coastal and Marine Science Center, St. Petersburg, FL 33701, USA;
jmiselis@usgs.gov (J.L.M.); nplant@usgs.gov (N.G.P.)
**\*** Correspondence: jbernier@usgs.gov

**Abstract:** The magnitude and frequency of storm events, relative sea-level rise (RSLR), sediment supply, and anthropogenic alterations drive the morphologic evolution of barrier island systems, although the relative importance of any one driver will vary with the spatial and temporal scales considered. To explore the relative contributions of storms and human alterations to sediment supply on decadal changes in barrier landscapes, we applied Otsu's thresholding method to multiple satellite-derived spectral indices for coastal land-cover classification and analyzed Landsat satellite imagery to quantify changes to the northern Chandeleur Islands barrier system since 1984. This high temporal-resolution dataset shows decadal-scale land-cover oscillations related to storm–recovery cycles, suggesting that shorter and (or) less resolved time series are biased toward storm impacts and may significantly overpredict land-loss rates and the timing of barrier morphologic state changes. We demonstrate that, historically, vegetation extent and persistence were the dominant controls on alongshore-variable landscape response and recovery following storms, and are even more important than human-mediated sediment input. As a result of extensive vegetation losses over the past few decades, however, the northern Chandeleur Islands are transitioning to a new morphologic state in which the landscape is dominated by intertidal environments, indicating reduced resilience to future storms and possibly rapid transitions in morphologic state with increasing rates of RSLR.

**Keywords:** Landsat; landscape evolution; barrier island; land cover; spectral indices; Otsu thresholding

## 1. Introduction

Prevailing oceanographic climate, sediment supply, relative sea-level rise (RSLR), the magnitude and frequency of storm events, and anthropogenic modifications interact to drive the morphologic evolution of barrier systems at varying spatial and temporal scales. Understanding the decadal- to centennial-scale evolution of barrier systems is critical for modeling future barrier change [1]. Whereas recent studies have applied improved methods and technologies to advance understanding of centennial-scale barrier evolution [2–7] or explore geologic controls on decadal-scale behavior [8–13], measuring decadal-scale barrier change is limited to observations from the historical record.

Historical analyses commonly focus on changes to the seaward-facing shoreline position or beach width and dune morphology [9,14–23]. Less frequently, changes to the whole-island area are also considered [15,17,21,24–26]. These studies often emphasize short-term changes induced by extreme storm events [14,15,23,27–29] and most consider the sandy barrier-island (beach and dune) and back-barrier (marsh and tidal flat) environments separately. Recent studies, however, demonstrated the importance of whole-system connectivity to barrier morphology and evolution [30–32] and expanded the scope of historical analyses to consider the annual- to decadal-scale landscape evolution of barrier islands [33–39].

To quantify historical change, researchers often must integrate diverse data sources of differing ages and with varying collection frequencies. Consequently, change analyses rely on varying methods including reinterpretation of irregularly updated historical map products that span multiple decades [15,31,37], photointerpretation of historical panchromatic or natural-color aerial imagery with decadal or longer time gaps between acquisition dates [15,40], and (or) classification of recent natural-color, color-infrared, or multispectral aerial and satellite imagery collected at sub-annual to interannual time scales [15,31,33,34,37,41]. The varying temporal and spatial resolution of these data sources make it difficult to compare results from different analyses.

The complexity of documenting and understanding historical barrier evolution is illustrated by case studies at the Chandeleur Islands, a transgressive deltaic barrier-island system [42–44] located about 40 km east of the Louisiana mainland in the northern Gulf of Mexico (Figure 1). Previous research documented severe shoreline erosion and land loss caused by Hurricane Katrina in 2005 [15,27], post-1996 decadal-scale habitat changes [39,45], and annual- to decadal-scale land-area changes and vegetative-productivity trends resulting from redistribution of sediment emplaced in 2010 [37,41]. Collectively, these studies addressed short-term land-area and land-cover changes after extreme storms and sediment placement separately, making it difficult to determine whether recent changes are the result of natural storm–recovery cycles or the addition of new sediment.

The goal of this study is to provide a comprehensive analysis of recent landscape-scale changes along the northern Chandeleur Islands using a consistent dataset and methodology to better understand temporal and spatial variability in barrier response to natural and anthropogenic disturbances over the past few decades, when we expect other drivers of barrier evolution (e.g., changes in oceanographic climate or SLR) are minimized. Despite the increasing availability of high-resolution multispectral satellite imagery in the last decade, medium-resolution (30-m pixel size) Landsat Thematic Mapper (TM; Landsat 5 mission), Enhanced Thematic Mapper Plus (ETM+; Landsat 7) and Operational Land Imaging (OLI; Landsat 8) imagery continue to be used for local- to global-scale land-cover classification and change analyses [35,46–53] since the imagery is freely available and provides a consistent, long-term (since 1984), high temporal resolution (16-day repeat cycle) and multispectral data source. Recently, automated processing of Landsat data has also been applied to extract satellite-derived shoreline positions and assess shoreline- and morphologic-change trends [54–58].

We analyzed Landsat satellite imagery from the northern Chandeleur Islands to quantify changes to subaerial and intertidal barrier environments resulting from storm impacts and human-induced sediment input at intra-annual to decadal time scales since 1984. Satellite-derived land-cover metrics were used to explore (1) the extent to which persistent back-barrier vegetation (marsh) influenced both "instantaneous" and longer-term morphologic change and recovery following storm events and (2) whether the addition of significant volumes of sediment to a sediment-starved system promoted greater island stability and (or) vegetative growth. The methods presented demonstrate the applicability of automatic thresholding techniques with multiple satellite-derived spectral indices for coastal land-cover classification and extraction of barrier metrics. Our results document historical changes at the northern Chandeleur Islands at higher temporal resolution than previous studies, offering new insight into alongshore variability of and morphologic controls on barrier-system evolution and providing additional metrics that complement recent sedimentologic [59], geophysical [60–62], and morphologic [63] analyses at the northern Chandeleur Islands. Furthermore, our analyses provide insight into drivers of barrier-island evolution over decadal time scales and how that might influence future vulnerability to storm events and SLR. The results also have important implications for resiliency of barrier islands, particularly those that are sediment-starved or exposed to frequent storm events.

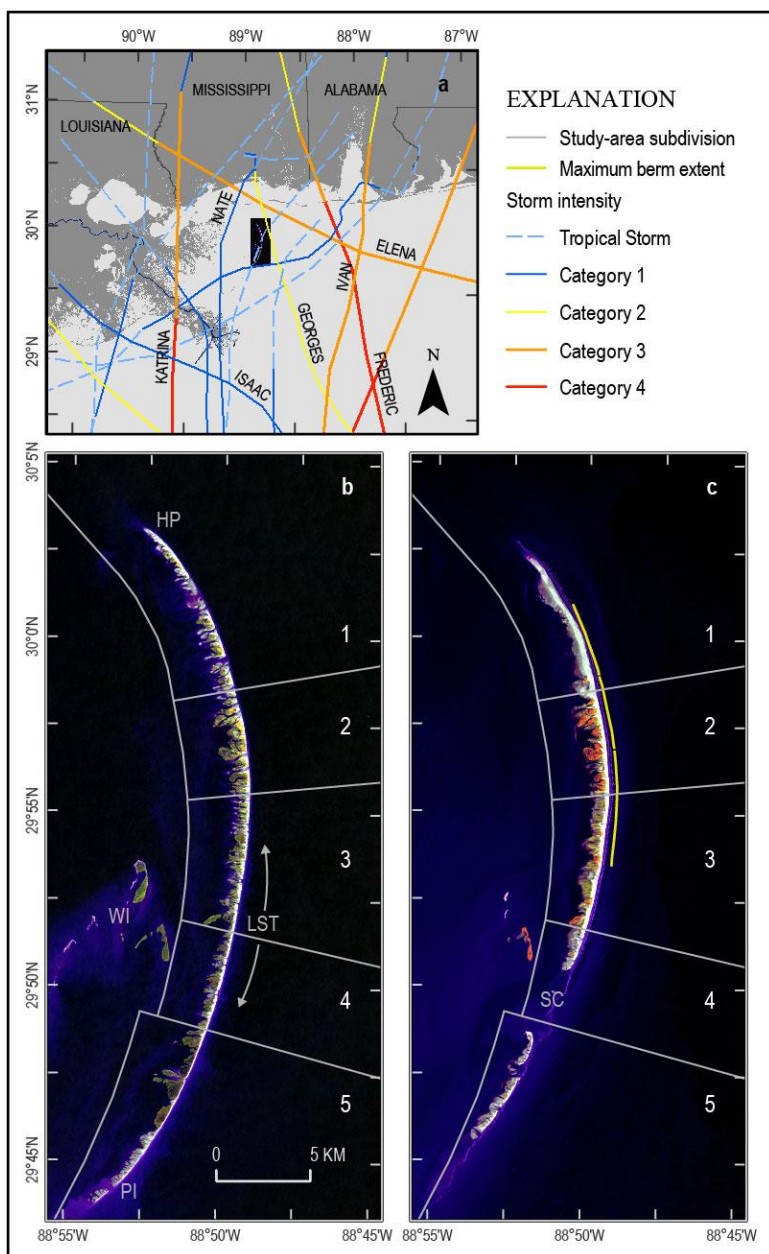

**Figure 1.** (**a**) Regional map showing locations of tropical cyclones that passed within 200 km of the study area since 1984. Hurricane Frederic (1979) is also shown. Inset satellite image indicates study-area extent shown in panels (**b**,**c**). (**b**) Landsat 5 satellite image acquired 25-March-1984 and (**c**) Landsat 8 satellite image acquired 10-January-2019 show subaerial configuration of the northern Chandeleur Islands at the beginning and end of the analysis period, respectively. Imagery is overlaid with study-area subdivisions and maximum as-constructed berm extent. False-color images use bands 4, 5, 3 (Landsat 5) or 5, 6, 3 (Landsat 8). [Abbreviations: HP, Hewes Point; LST, approximate latitude of longshore transport node; PI, Palos Island; SC, Smack Channel Cut; WI, Western Islands].

## 2. Materials and Methods

### 2.1. Study Area

Part of the Breton National Wildlife Refuge, the Chandeleur Islands provide a critical habitat for migratory and nesting birds, are an economically important recreational fishery [64], maintain estuarine conditions in Breton and Chandeleur Sounds and adjacent mainland wetlands [65], and provide protection to coastal wetlands and human infrastructure from both fair-weather and storm-induced waves [66]. In recent decades, the

Chandeleur Islands narrowed significantly or, as at Curlew and Grand Gosier Islands, were reduced to ephemeral shoals. Consequently, these islands have become increasingly vulnerable to both fair-weather and storm-induced land loss [15,43,44,67]. The transgressive submergence of the Chandeleur Islands is largely the result of alongshore processes that transport much of the sediment eroded from the shoreface to northern and southern deep-water depositional centers, limiting sediment availability for island roll-over and back-barrier land building through overwash deposition [44,67–70]. With the exception of a short-lived fishing settlement in the early 1900s [64] and a 2001 effort to stabilize unvegetated washover deposits that formed during Hurricane Georges (1998) using plantings of *Spartina alterniflora* at 10 sites [71], the Chandeleur Islands historically were largely unaltered by human activities. Between June 2010 and March 2011, however, the State of Louisiana constructed a 2-m high sand berm extending more than 14 km along the northern Chandeleur Islands barrier platform (Figure 1) as part of emergency response efforts to prevent oiling from the Deepwater Horizon oil spill [37,63,72].

The study area encompasses the northern ~38 km of the Chandeleur Islands from Hewes Point to Palos Island (Figure 1) and includes the modeled longshore transport (LST) nodal point [68,69]. The western islands (New Harbor, North, and Freemason Islands), which are not subject to the same open-oceanographic forcing as the fronting barrier-island arc, were excluded from this analysis. The northern Chandeleur Islands historically formed a semi-continuous island [66] characterized by small, mostly ephemeral inlets that formed during storm events but usually re-closed during intervening fair-weather periods [43,73]. Between 1979 and 2019, 17 tropical systems passed within 200 km of the study area (Figure 1, Table S1), including Hurricanes Frederic (1979), Elena (1985), Georges (1998), Ivan (2004), Katrina (2005), and Isaac (2012). Hurricanes Frederic, Elena, and Georges caused beach erosion, breaching, and overwash deposition along the northern Chandeleur Islands [15,73]. Hurricanes Ivan and Katrina opened more than 60 inlets along the northern islands [67,71], many of which remained open in 2010 prior to berm construction [43]. Although a weaker system than these preceding storms, Hurricane Isaac impacted the study area only a year after berm construction was completed and caused significant offshore-directed cross-shore sediment transport [74]. The islands are extremely low-lying, with 2005 to 2010 (pre-berm) mean emergent-island elevations ranging from about 0.33 to 0.42 m NAVD88 (0.17 to 0.26 m above mean sea level [MSL]) and maximum (dune) elevations during the same period varying from about 1.35 to 2.39 m NAVD88 (1.19 to 2.23 m MSL) [37]. The study area is microtidal, with a mean tidal range of 0.42 m. Emergent habitats are dominated by sandy beaches; low, hummocky dunes; sparsely vegetated washover fans; and emergent salt marshes [37,71]. To assess alongshore variability in the evolution of these islands, we divided the study area into five subset areas based on qualitative observations of island morphology (Figure 1, Table 1).

**Table 1.** Morphologic subdivision of northern Chandeleur Islands study area.

| Subarea | Description |
|---|---|
| 1 | Fragmented post-Hurricane Georges marsh platform (berm built on submerged barrier platform) |
| 2 | Persistent post-Hurricane Katrina marsh platform (berm built in front of or attached to emergent island) |
| 3 | Fragmented post-Hurricane Katrina marsh platform, north of longshore-transport (LST) nodal point (berm, if present, built attached to emergent island) |
| 4 | Persistent post-Hurricane Georges marsh platform, south of LST nodal point (south of berm emplacement) |
| 5 | Fragmented post-Hurricane Georges marsh platform, south of post-Hurricane Katrina Smack Channel Cut |

### 2.2. Data Sources and Image Pre-Processing

One hundred and ninety-three Landsat TM, ETM+, and OLI images (Worldwide Reference System 2 [WRS-2] path 21 row 39) acquired between March 1984 and January 2019 were identified that were cloud free and either pre-scan line corrector failure (SLC-off) or SLC-off gap free (Landsat 7 ETM+) over the study area. To minimize the effects of water-level variations on land-cover classification [75,76], only images that were collected

within 2 h of predicted low tide or were collected on a rising tide with predicted water levels at time of acquisition less than MSL (National Ocean Service [NOS] Center for Operational Oceanographic Products and Services [CO-OPS] station 8760172, Chandeleur Light, LA [77]) were analyzed. The resulting dataset consisted of 52 Landsat 5 TM, 10 Landsat 7 ETM+, and 13 Landsat 8 OLI images.

For each image acquisition date, top-of-atmosphere (TOA) reflectance (reflective bands), TOA brightness temperature (BT; thermal infrared [TIR] bands), and surface reflectance-derived normalized difference vegetation index (NDVI) images were downloaded from the U.S. Geological Survey (USGS) Earth Resources Observation and Science (EROS) Center Science Processing Architecture (ESPA) On Demand Interface [78]. NDVI, defined as the normalized ratio between the near-infrared (NIR) and visible red (R) bands [79,80]:

$$NDVI = (NIR - R)/(NIR + R), \tag{1}$$

is widely accepted as a measure of vegetation "greenness" and is used both to indicate presence or absence of vegetated areas [37,81,82] as well as an indicator of vegetation health or composition [37,52,76,82,83].

All images were batch-processed using Spatial Model Editor in ERDAS IMAGINE® 2016. The TOA and BT bands (Table S2) were stacked to create 7- (TM, ETM+) or 9- (OLI) band multispectral images and clipped to the study-area extent. Because most of the images used are Landsat 5 TM products, which do not include a panchromatic band, images were not pan-sharpened during processing. From these composite images, two additional spectral indices, the modified normalized difference water index (mNDWI) [84] and the normalized difference bare land index (NBLI) [85] were calculated, defined as the normalized ratio between the visible green (G) and shortwave-infrared (SWIR) or visible red and TIR bands, respectively:

$$mNDWI = (G - SWIR_1)/(G + SWIR_1), \tag{2}$$

$$NBLI = (R - TIR)/(R + TIR). \tag{3}$$

mNDWI has been applied to differentiate between land and water areas at local [57] to regional ([48] scales and was selected for use in this study over the normalized difference water index (NDWI) [86] because of its previously demonstrated utility at distinguishing between land and water along Louisiana's wetland-dominated coastal regions [48]. Originally developed to differentiate between bare land and other land-cover types, especially built-up areas, in urban settings, Li et al. (2017) [85] demonstrated that the use of the TIR band in NBLI also provides good separation of bare land from water and vegetated areas.

*2.3. Land-Cover Classification and Feature Extraction*

Land-cover classification was performed using successive thresholding of spectral indices. First proposed by Kuleli et al. (2011) [87], automatic thresholding of water indices using Otsu's method [88] has been widely applied to segment satellite images into land and water areas and delineate shorelines [54–57] as well as for vegetation mapping [83,89]. Estoque and Murayama [90] applied Otsu thresholding to create a binary land-water image for the purpose of masking water areas from built-up indices. They then applied Otsu thresholding to the masked built-up datasets to further map the land area into built-up and non-built-up classes.

We developed a similar workflow to map four land-cover classes (water, bare earth [sand], vegetated, and intertidal) at the northern Chandeleur Islands (Figure 2a): first, Otsu's method was applied to the mNDWI image to create a binary land-water raster for each image acquisition date (Figure 2b). Second, water area was masked from the NBLI and NDVI images, and Otsu's method was applied to the masked images to create binary sand–"unclassed" and vegetated–"unclassed" rasters (Figure 2c), respectively. Because thresholding was applied to masked NBLI and NDVI simultaneously, some pixels along the sand-vegetation boundary were classed as both sand and vegetated in this step. We

mapped these pixels as sand using the following rule: if NBLI = sand and NDVI = vegetated, then final = sand. Next, water, sand, and vegetated areas were masked from mNDWI images with the remaining pixels classed as intertidal, and Otsu's method was applied to the masked mNDWI image to further separate "submerged" and "emergent" intertidal subclasses (Figure 2d). Finally, the binary land-cover images were converted to thematic rasters, merged, and single-pixel "clumps" were removed using a 3 × 3 majority filter to create a final land-cover raster dataset. All steps were batch-processed using the Image Processing toolbox in MATLAB® version R2018a (Otsu thresholding and binary image creation) or Spatial Model Editor in ERDAS IMAGINE® 2016 (spectral index masking and generation of classed land-cover rasters). For 15 datasets, thresholding resulted in misclassification of back-barrier seagrass beds as vegetated. These were identified based on visual comparison with ancillary datasets as well as vegetation persistence maps and misclassed extents were manually cleaned from the final datasets.

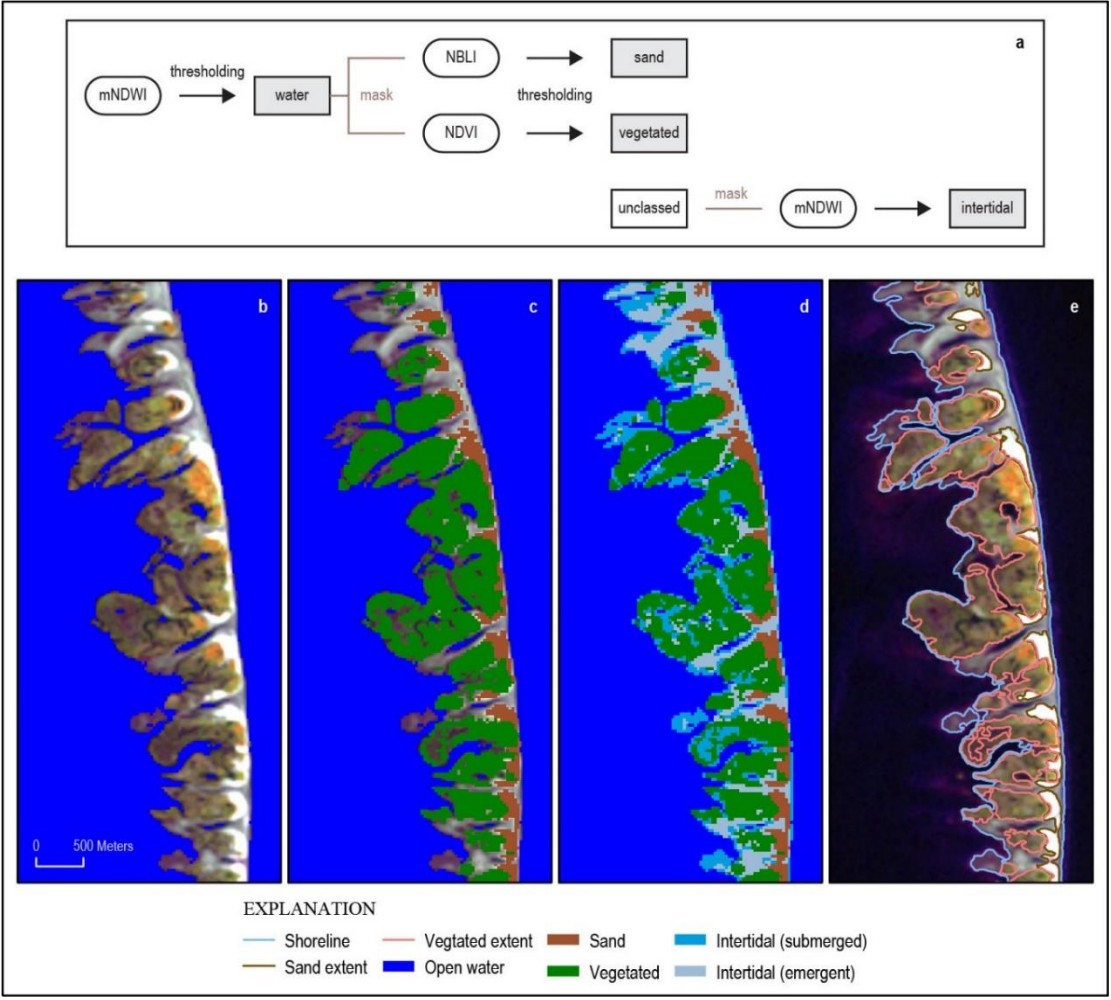

**Figure 2.** (**a**) Land-cover classification workflow, showing successive thresholding and masking of modified normalized difference water index (mNDWI), normalized difference bare land index (NBLI), and normalized difference vegetation index (NDVI) spectral indices to extract (**b**) water, (**c**) sand and vegetated, and (**d**) intertidal subclasses defined in this study. (**e**) Vector shoreline, sand, and vegetated extents were extracted by contouring the mNDWI, NBLI, and NDVI images using the calculated Otsu thresholds. Extent shown in panels (**b**–**e**) corresponds to subarea 2 (Figure 1; Table 1) and uses Landsat 5 image acquired 31-January-1986 as example.

Accuracy assessments were performed for 10 classed land-cover datasets from this study by comparing them to temporally similar Louisiana Barrier Island Comprehensive Monitoring Program (BICM) [39,45,91] or National Land Cover Dataset (NLCD) [50,92]

classed data (Table S3). To facilitate the analysis, reference land-cover types were reclassed to vegetated and bare earth (sand) classes consistent with the land-cover types defined in this study. The overall, user's (errors of omission: the probability that a classed land-cover pixel correctly matches the reference data), and producer's (errors of commission: how well the reference data are mapped in the classed data) accuracy for each dataset were computed from confusion matrices in Esri ArcGIS 10.5.

Vector barrier platform (seaward shoreline and landward limit of the barrier platform, including back-barrier intertidal areas), sand, and vegetated extents (Figure 2e) were extracted by contouring the mNDWI and masked NBLI and NDVI images, respectively, using the calculated Otsu thresholds. Contouring was batch-processed using PythonWin 2.7 with Esri ArcGIS 10.5. Vector features corresponding to a minimum mapping unit of less than five pixels were excluded. The classed land-cover datasets, vector feature extents, and complete metadata are available for download from [93].

### 2.4. Change Analyses

Areal land-cover changes calculated from the classed raster datasets were evaluated across the entire study area as well as the subset areas (Figure 1, Table 1). Additionally, for each land-cover dataset, barrier platform, beach, and vegetated (marsh) widths were calculated from the intersection of the shoreline, sand, and vegetated vectors with transects spaced 300 m (10 pixels) apart alongshore (Figure S1). The software package AMBUR (Analyzing Moving Boundaries Using R) [94] was used to extract feature positions along the transects as well as to calculate shoreline-change rates for the seaward barrier shoreline.

Vegetation persistence was evaluated by reclassifying each land-cover raster such that vegetated pixels were assigned a value of 1 and all other pixels were assigned a value of 0. The reclassed rasters were then summed, and the pixel values of the resulting raster represent the number of images for which each pixel was classed as vegetated. Raster processing was performed using Spatial Model Editor in ERDAS IMAGINE® 2016.

### 3. Results

#### 3.1. Classification Accuracy

Overall classification accuracy comparing the water, bare earth (sand), and vegetated classes from land-cover datasets generated in this study with temporally consistent NLCD and BICM datasets ranged from 50% to 80% (Tables 2 and S4) and was greater than about 70% for 10 of the 14 assessments, which is comparable to classification accuracies reported for NLCD datasets [50]. For these data, the user's accuracy was consistently greatest for water (>93%). User's accuracy for the sand and vegetated classes was more variable, and errors of commission usually occurred when water areas in the reference datasets were represented by sand or vegetated pixels in the classed images. Less commonly, reference vegetated extents were represented by sand pixels in the classed data.

**Table 2.** Overall classification accuracy between classed land-cover data from this study and reference datasets comparing water, bare earth (sand), vegetated, and intertidal classes at the northern Chandeleur Islands. [Abbreviations: NLCD, National Land Cover Dataset; BICM, Barrier Island Comprehensive Monitoring program; N/A, not applicable].

| Classed Data | Reference Dataset (NLCD) | Overall Accuracy [1] | Reference Dataset (BICM) | Overall Accuracy [1] | Overall Accuracy [2] |
|---|---|---|---|---|---|
| 17-November-2016 | NLCD 2016 | 69.3% | BICM 2016 | 74.0% | 61.8% |
| 18-January-2016 | NLCD 2016 | 80.3% | BICM 2016 | 71.0% | 61.0% |
| 24-October-2013 | NLCD 2013 | 71.7% | N/A | N/A | N/A |
| 03-October-2011 | NLCD 2011 | 78.3% | N/A | N/A | N/A |
| 30-January-2009 | NLCD 2008 | 55.7% | BICM 2008 | 70.3% | 66.5% |
| 18-October-2005 | NLCD 2006 | 51.0% | BICM 2005 | 50.0% | 40.8% |
| 15-October-2004 | NLCD 2004 | 57.3% | N/A | N/A | N/A |
| 31-March-2004 | N/A | N/A | BICM 2004 | 75.3% | 63% |
| 21-September-2001 | NLCD 2001 | 71.0% | N/A | N/A | N/A |
| 17-Febuary-1998 | N/A | N/A | BICM 1998 | 74.3% | 56.5% |

[1] Compares water bare earth (sand), and vegetated classes only. [2] Compares water, bare earth (sand), vegetated, and intertidal classes against BICM reference data only.

Overall classification accuracies less than 60% (Tables 2 and S4) were calculated for four assessments: comparison of classed images acquired immediately post-hurricanes Ivan (15-October-2004 Landsat 5 with NLCD 2004 reference data) and Katrina (18-October-2005 Landsat 5 with NLCD 2006 and BICM 2005 reference data) and comparison of a classed image acquired 30-January-2009 with NLCD 2008 reference data. Confusion matrices (Table S4) for these datasets indicate that extremely low user's accuracy (0–34%) for the sand class contributed significantly to the poorer overall classification accuracy.

NLCD reference datasets map subaerial land cover only, however, the BICM reference datasets also mapped intertidal extents. Overall classification accuracies for six datasets comparing water, sand, vegetated, and intertidal classes with BICM reference data (Table S5) were about 11% lower than comparisons between the same datasets that excluded intertidal areas (Table 2). The predominant errors driving this varied, and we note that both water levels and water clarity affect the spectral characteristics of the images, and, therefore, the spectral differences that separate water from intertidal extents. Additionally, classification methods differed among the reference datasets (e.g., BICM 1998, 2004, and 2005 [45] vs. BICM 2008 and 2016 [39]). None of the reference datasets allow for ground-truthing of the intertidal "submerged" and intertidal "emergent" subclasses. However, comparison with lidar elevations (Figure S2) shows that for each of the datasets analyzed, intertidal "submerged" elevations were significantly different ($p < 0.05$) than intertidal "emergent" elevations, supporting the separation of these subclasses.

### 3.2. Temporal Land-Cover Changes

From a whole-island perspective, significant decreases ($p < 0.01$; Table S6) in sand, vegetated, total island (sand plus vegetated), and barrier platform (sand plus vegetated plus intertidal) extents (Figure 3a,b) and percent vegetated cover (Figure 3c) were observed after hurricanes Georges (28-September-1998) and Katrina (29-August-2005) impacted the study area. Dividing the analysis extent into subareas (Table 1) allowed us to explore variability in that response and revealed that the magnitude and timing of land-cover changes varied alongshore (Figures 4 and 5). For example, post-Georges vegetation losses in the central study area (subareas 2 and 3; Figure 4b,c) were less extensive than elsewhere, barrier-platform area was relatively stable, and vegetated extents remained the dominant land-cover type (Figure 5b,c). In the northern and southern study area, intertidal extents became increasingly important to the barrier-platform area (Figure 5a,d,e). Subareas 1 and 5 experienced post-Georges vegetation losses that contributed to observed decreases in barrier-platform area (Figure 4a,e). Immediately south of the longshore transport (LST) node, barrier-platform area in subarea 4 (Figure 4d) was relatively stable until 2005, but percent vegetated cover decreased significantly following Hurricane Georges (Figure 5d). Together, these observations suggest the onset of a transition from emergent vegetation to unvegetated back-barrier tidal flats across much of the study area. Following Hurricane Katrina, vegetation, although reduced in total extent, remained the dominant land-cover type only in subarea 2 (Figures 4b and 5b). In the northern study area (subarea 1) and south of the LST node (subareas 4 and 5), the barrier platform was mostly submerged following Hurricane Katrina, and land cover was dominated by tidal flats.

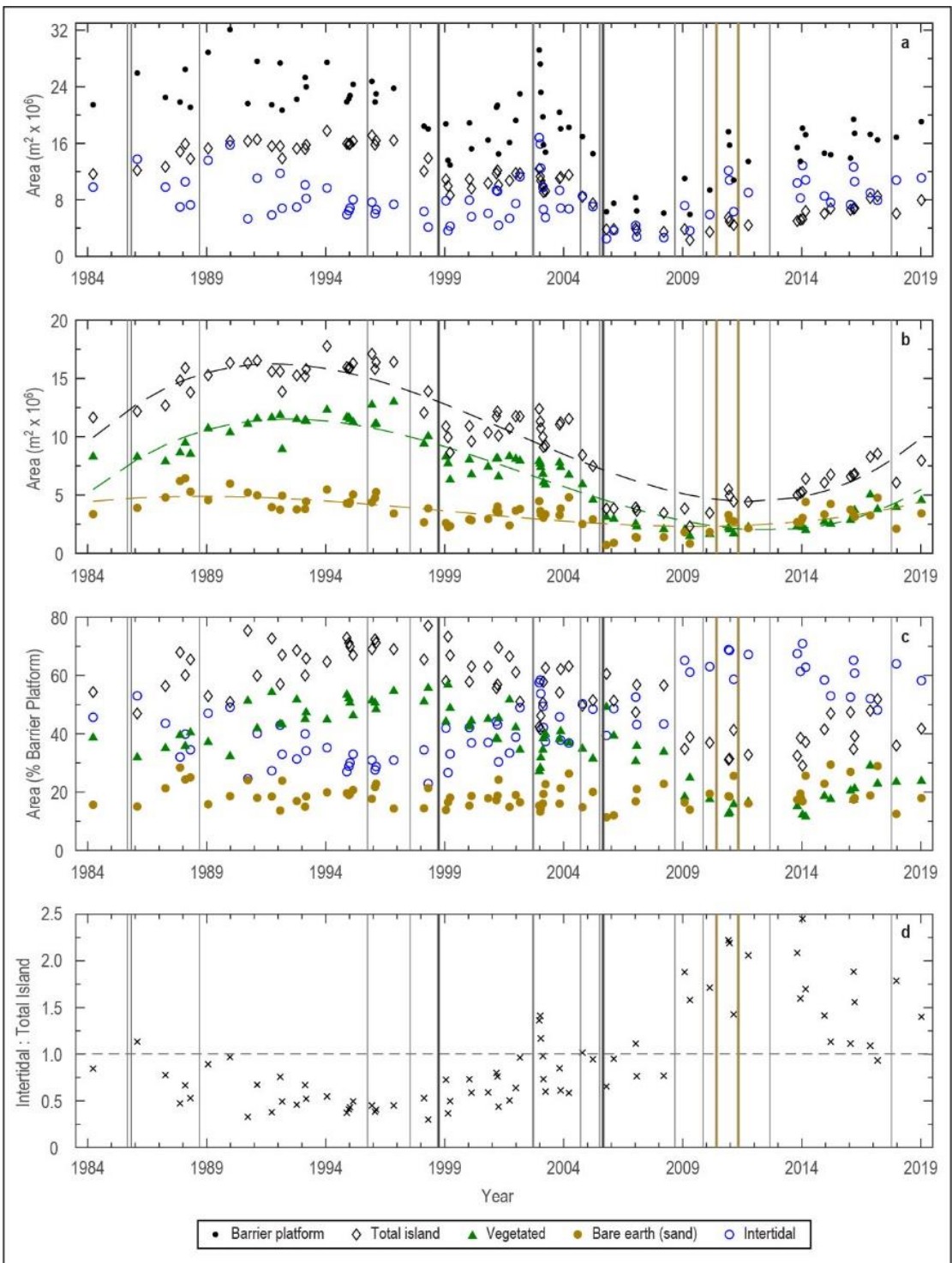

**Figure 3.** Temporal changes in (**a**,**b**) land-cover area, (**c**) land-cover extent as a percent of total barrier-platform (sand plus vegetated plus intertidal) area, and (**d**) ratio of intertidal to total island (sand plus vegetated) extents across the entire study area. After 2010, the barrier platform was dominated by intertidal areas, indicated by values in (**d**) that are greater than 1 (horizontal dashed line). The timing of berm construction (brown vertical lines) and tropical cyclones (gray vertical lines) that passed within 200 km of the northern Chandeleur Islands are shown. Significant decreases in sand, vegetated, total island (sand plus vegetated), and barrier platform (sand plus vegetated plus intertidal) extents were observed after Hurricanes Georges (28-September-1998) and Katrina (29-August-2005) (black vertical lines); long-term changes show somewhat cyclical trends, indicated by dashed lines in (**b**), that can be related to storm–recovery cycles.

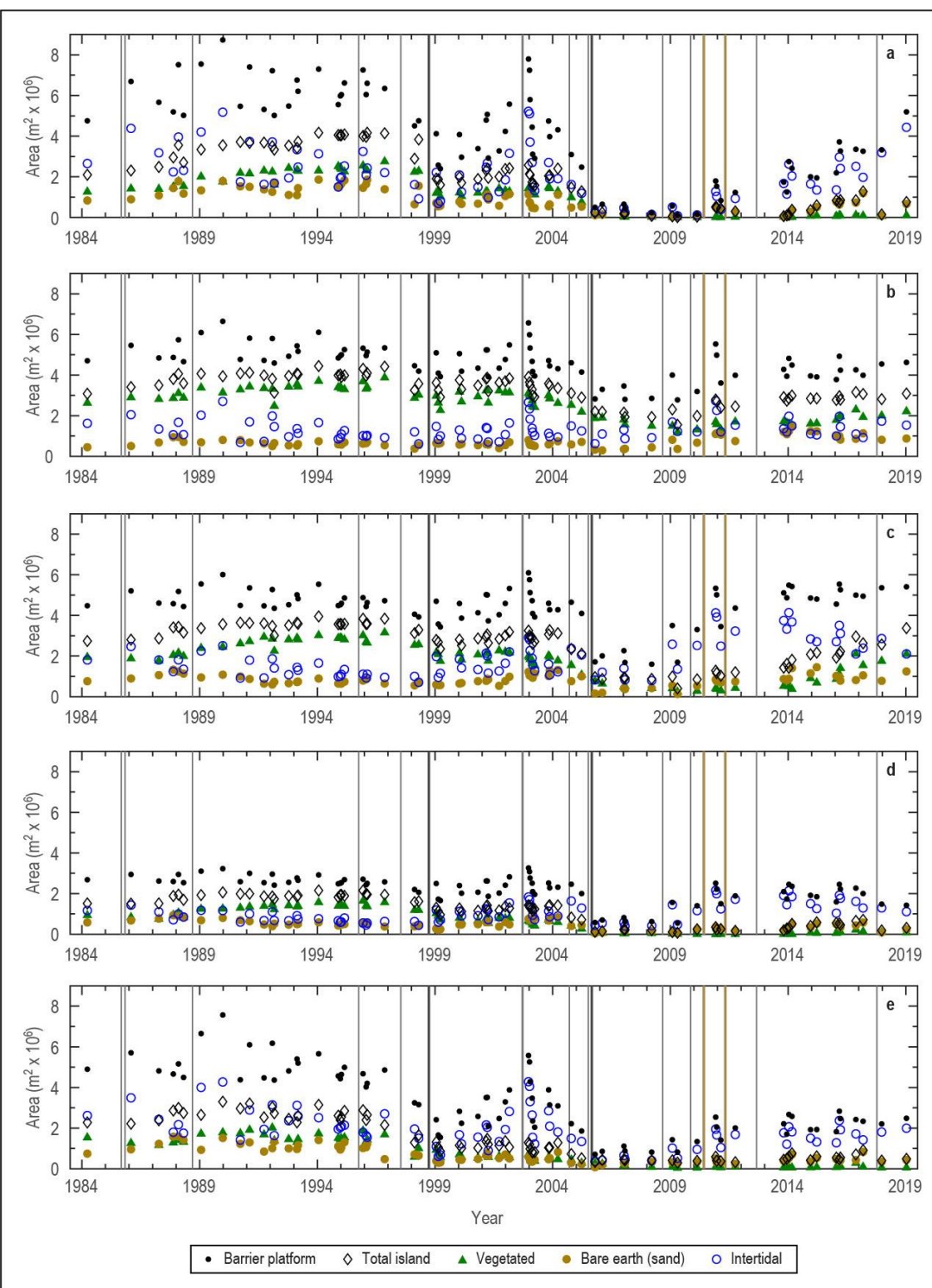

**Figure 4.** Temporal changes in land-cover extents for (**a**) subset area 1, (**b**) subset area 2, (**c**) subset area 3, (**d**) subset area 4, and (**e**) subset area 5 illustrates alongshore-variable response of the northern Chandeleur Islands to perturbations such as tropical cyclones (gray vertical lines), including Hurricanes Georges (28-September-1998) and Katrina (29-August-2005) (black vertical lines), and berm construction (brown vertical lines). See Table 1 for description of study area subdivisions.

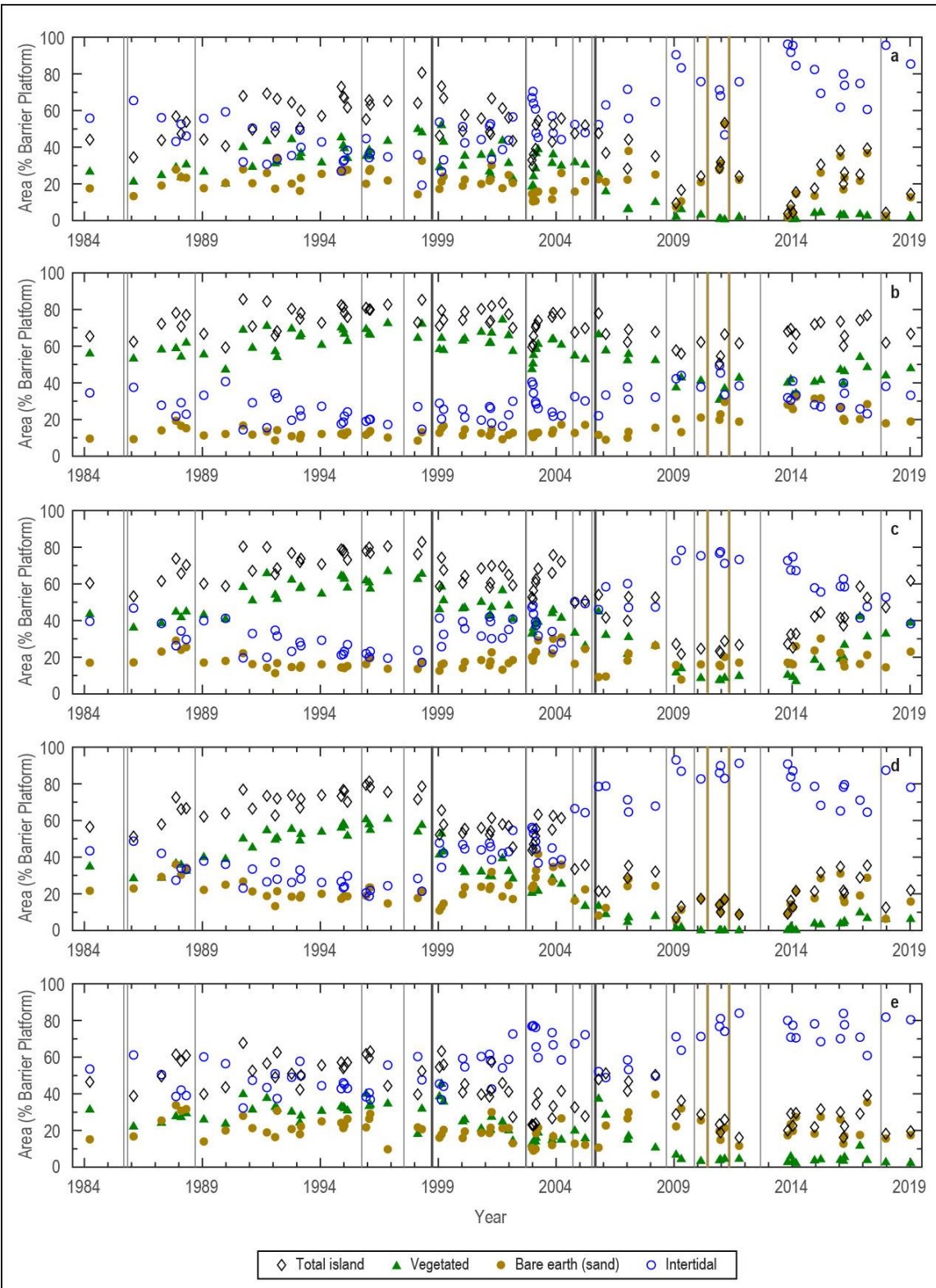

**Figure 5.** Temporal changes in land-cover extents as a percent of total barrier-platform (sand plus vegetated plus intertidal) area at the time of image acquisition for (**a**) subset area 1, (**b**) subset area 2, (**c**) subset area 3, (**d**) subset area 4, and (**e**) subset area 5 illustrates alongshore-variable response of the northern Chandeleur Islands to perturbations such as tropical cyclones (gray vertical lines), including Hurricanes Georges (28-September-1998) and Katrina (29-August-2005) (black vertical lines), and berm construction (brown vertical lines). See Table 1 for description of study area subdivisions.

Since 2010 and following berm construction, increases in barrier-platform area were observed along the length of the barrier-island chain (Figure 4). In subareas 1, 4, and 5, these gains reflect increases in sandy and intertidal extents with minimal changes in post-Katrina vegetated extents. In the central study area (subareas 2 and 3), increases in both total and percent vegetation cover were observed (Figures 4b,c and 5b,c), although increases in intertidal extents contributed more than vegetation to gains in subarea 3 (Figures 4c and 5c). Except for subarea 2, the post-2010 landscape was dominated by intertidal areas (Figure 3d).

Land-cover changes can be related to storm–recovery cycles (Figure 3b): between 1984 and ~1989, vegetated and total island extents increased, which we interpret as recovery following Hurricanes Frederic and Elena. Between 1989 and April 1998, when storms were infrequent, total island area stabilized. Following Hurricane Georges, total island and barrier-platform areas decreased, largely caused by declines in vegetated extent. Significant decreases in all land-cover extents were observed beginning in October 2005, immediately after Hurricane Katrina. Minimal recovery was observed until 2010, when both sandy and vegetated extents began to increase. Abrupt increases in sandy extents, especially in subarea 1 (Figures 4a and 5a), can be attributed to berm construction. Post-2010 vegetated extents increased gradually, similar to post-storm recovery trends observed between 1984 and 1989, although vegetated and total island extents have not recovered to pre-Georges levels.

### 3.3. Barrier Metrics

Barrier metrics (feature positions and widths) were evaluated for four time periods (Table 3), corresponding to natural (storm) and anthropogenic (berm construction) events that affected historical land-cover changes (Figures 2–4). Figure 6 shows the distribution of the back-barrier platform and sea shoreline positions along each transect relative to the offshore baseline and 1984 barrier footprint (Figure S1). The barrier platform retreated landward along the length of the northern Chandeleur Islands during the study period; however, landward translation of both the sea shoreline (Figure 6b) and the back-barrier platform extent (Figure 6a) increased in magnitude south of subarea 2, with the highest magnitudes of change south of the LST nodal point. For example, following Hurricane Georges, the sea shoreline (Figure 6b, brown bars) south of transect 24 in subarea 5 had retreated almost completely landward of the 1984 back-barrier platform extent, a trend that was observed along most of the length of the barrier-island chain south of the LST node (subareas 4 and 5) after Hurricane Katrina. In subarea 2, where vegetation remained the dominant land-cover even following Hurricane Katrina, the position of the back-barrier platform was relatively stable throughout the analysis period (Figure 6a). Sea shoreline-change rates (Table 4, Figure S3) show similar trends: the highest shoreline-erosion rates generally occurred south of the LST node (subareas 4 and 5) and shoreline-erosion rates increased dramatically following Hurricane Katrina throughout the study area except for subarea 2. Between 2010 and 2019, shoreline erosion occurred south of subarea 1, with particularly high rates south of the LST node (subareas 4 and 5). Apparent shoreline accretion associated with subaerial emergence within the historic (1984) subarea barrier-platform footprint during the same period likely reflects redistribution of berm sediment (Figures 6b and S3, blue bars).

**Table 3.** Analysis time periods based on natural and anthropogenic events that affected land-cover changes in the northern Chandeleur Islands, with the number of images analyzed per period.

| Time Period | Description | Number of Images |
| --- | --- | --- |
| 25-March-1984 to 22-April-1998 | Pre-Hurricane Georges [1] (14.1 years) | 27 |
| 10-January-1999 to 24-March-2005 | Pre-Hurricane Katrina [2] (6.2 years) | 23 |
| 18-October-2005 to 18-Febuary-2010 | Post-Hurricane Katrina [2] (4.3 years) | 8 |
| 3-December-2010 to 19-January-2019 | Post-berm construction [3] (8.1 years) | 17 |

[1] 28-September-1998; [2] 29-August-2005; [3] June-2010 to March-2011.

**Table 4.** Linear regression sea-shoreline change rates in m/y at the northern Chandeleur Islands averaged by subarea and analysis period, with outliers removed. The total number of transects in each subarea (Figure S1) is listed; 1σ "error" represents spatial variability in change rates among transects within each subarea.

| Subarea | Number of Transects | 1984–1998 | 1999–2005 | 2005–2010 | 2010–2019 |
|---|---|---|---|---|---|
| 1 | 33 | $-1.2 \pm 1.6$ | $1.9 \pm 19.7$ | $-115.8 \pm 60.3$ | $5.7 \pm 12.2$ |
| 2 | 19 | $-3.2 \pm 1.2$ | $-2.8 \pm 1.7$ | $7.8 \pm 20$ | $-11.2 \pm 4.8$ |
| 3 | 25 | $-3.6 \pm 1.1$ | $-1.1 \pm 5.9$ | $-16.8 \pm 38.6$ | $-9.8 \pm 5$ |
| 4 | 18 | $-5.5 \pm 1.8$ | $-7.4 \pm 7$ | $-58.6 \pm 61.7$ | $-31.5 \pm 14.3$ |
| 5 | 37 | $-13.9 \pm 4.3$ | $-25.6 \pm 23.6$ | $-116.2 \pm 18.4$ | $-45.5 \pm 24.3$ |

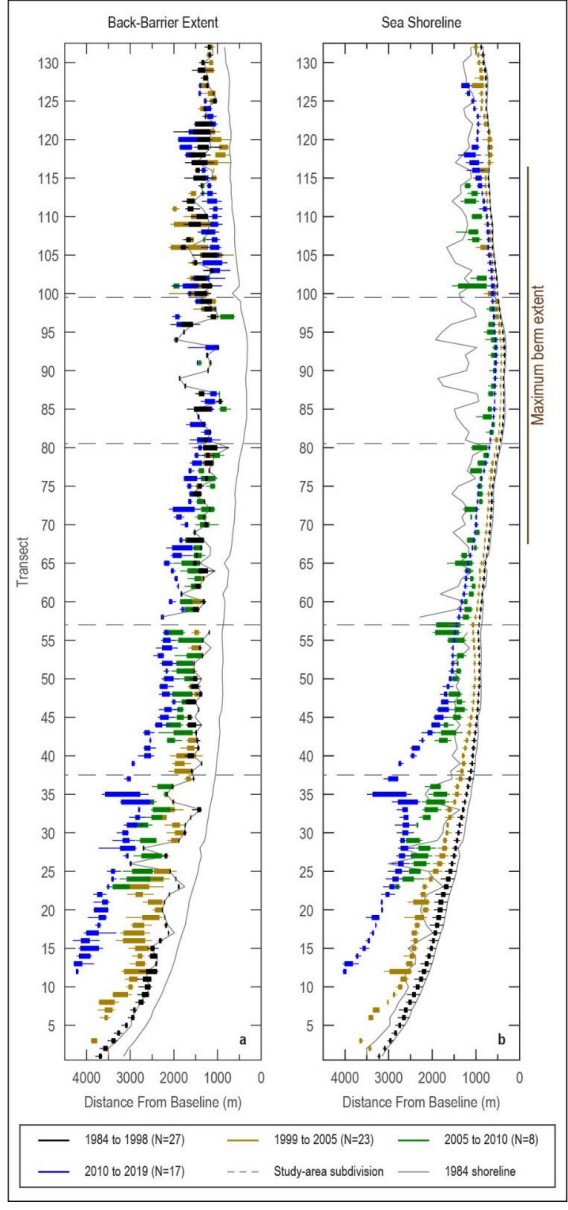

**Figure 6.** Boxplot showing distribution of (**a**) back-barrier platform and (**b**) sea-shoreline positions along cross-shore transects spaced 300 m alongshore relative to offshore baseline (Figure S1) at the northern Chandeleur Islands for 25-March-1984 to 22-April-1998 (pre-Hurricane Georges), 10-January-1999 to 24-March-2005 (pre-Hurricane Katrina), 18-October-2005 to 18-Febuary-2010 (post-Hurricane Katrina), and 3-December-2010 to 19-January-2019 (post-berm construction). 25-March-1984 barrier-platform extent is delineated by gray line; dashed horizontal lines indicate study area subdivision boundaries.

The average width of the barrier platform and vegetated extents along each transect are shown in Figure 7. Throughout the analysis period, barrier platform and vegetated widths in the central part of the study area (subarea 2) were relatively stable or narrowed only slightly. Elsewhere, the timing of narrowing or loss of the vegetated platform varied alongshore. Following Hurricane Georges, the formerly continuous back-barrier vegetation in the northern- and southern-most study area (subareas 1 and 5) became fragmented, and vegetation extents in subarea 4 narrowed relative to pre-Georges extents (Figure 7b). Following Hurricane Katrina, the subarea 3 vegetated platform became fragmented, remnant vegetation in subareas 1, 4, and 5 was mostly removed, and the barrier platform narrowed along most of the study area (Figure 7c).

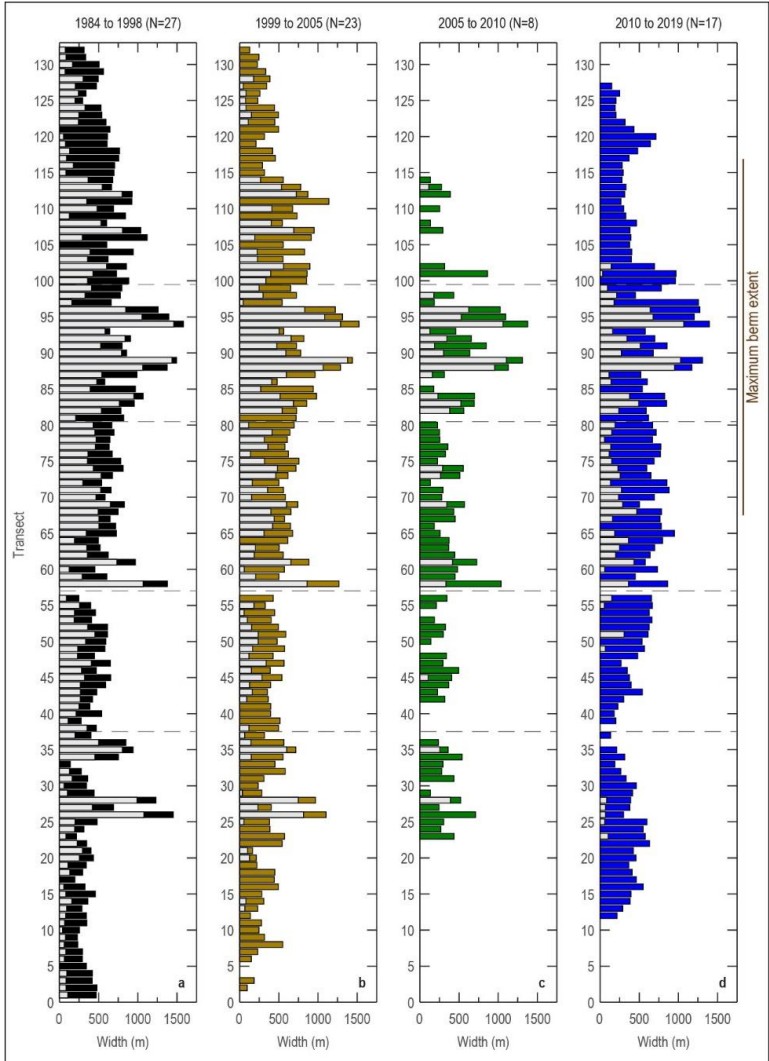

**Figure 7.** Plot showing barrier-platform (colored bars) and vegetated (gray bars) widths averaged along cross-shore transects spaced 300 m alongshore (Figure S1) at the northern Chandeleur Islands for (**a**) 25-March-1984 to 22-April-1998 (pre-Hurricane Georges), (**b**) 10-January-1999 to 24-March-2005 (pre-Hurricane Katrina), (**c**) 18-October-2005 to 18-Febuary-2010 (post Hurricane Katrina), and (**d**) 3-December-2010 to 19-January-2019 (post-berm construction). Bars represent average feature width based a minimum of 3 observations per transect per period; no bar indicates 0 (not present), 1, or 2 observations per transect per time period. Dashed horizontal lines indicate study area subdivision boundaries. Feature widths were not calculated along transect 57, where the oblique nature of the back-barrier marsh island, coupled with a change in baseline orientation, caused inconsistencies in delineating feature extents.

Notably, the smallest post-Katrina decreases in barrier-platform width occurred in subarea 2 (Figure 7c), where pre-Katrina (post-Georges) vegetated widths (Figure 7a) were higher. This is consistent with island-wide trends, in which the barrier platform is generally widest where the vegetated platform is also widest. In Figure 8, linear regression analysis shows that since 2010, barrier-platform and vegetated widths were not as strongly correlated as during earlier time periods, likely reflecting increased dominance of intertidal areas (Figures 3–5 ) and losses to the persistent vegetated (marsh) platform through time (discussed below and shown in Figures 9 and 10).

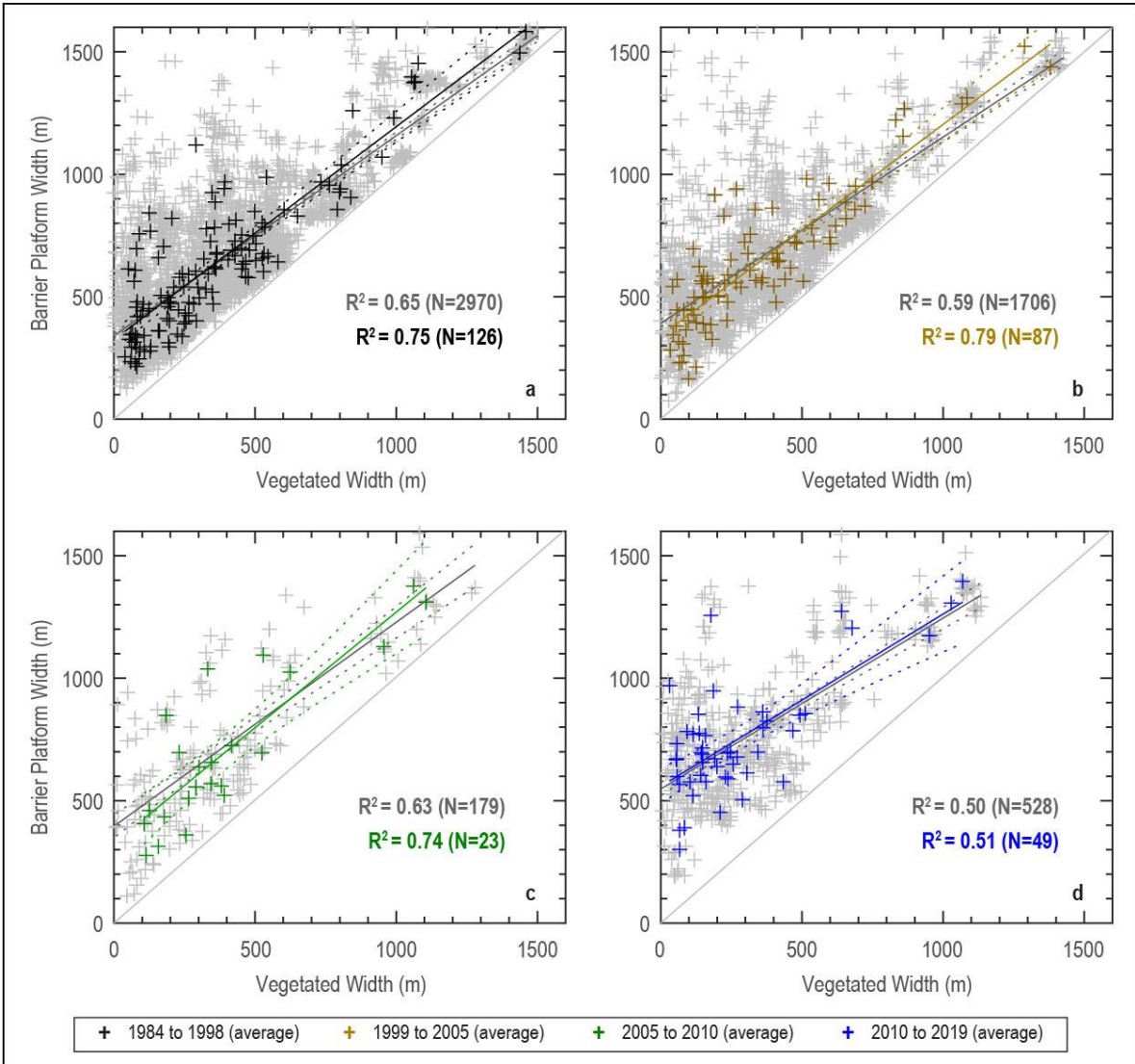

**Figure 8.** Linear regression plots of barrier-platform versus vegetated width along cross-shore transects spaced 300 m alongshore (Figure S1) at the northern Chandeleur Islands for (**a**) 25-March-1984 to 22-April-1998 (pre-Hurricane Georges), (**b**) 10-January-1999 to 24-March-2005 (pre-Hurricane Katrina), (**c**) 18-October-2005 to 18-Febuary-2010 (post Hurricane Katrina), and (**d**) 3-December-2010 to 19-January-2019 (post-berm construction). Individual observations (light gray crosses) are overlaid with values averaged along cross-shore transects for each period; dashed lines indicate regression 95% confidence bounds.

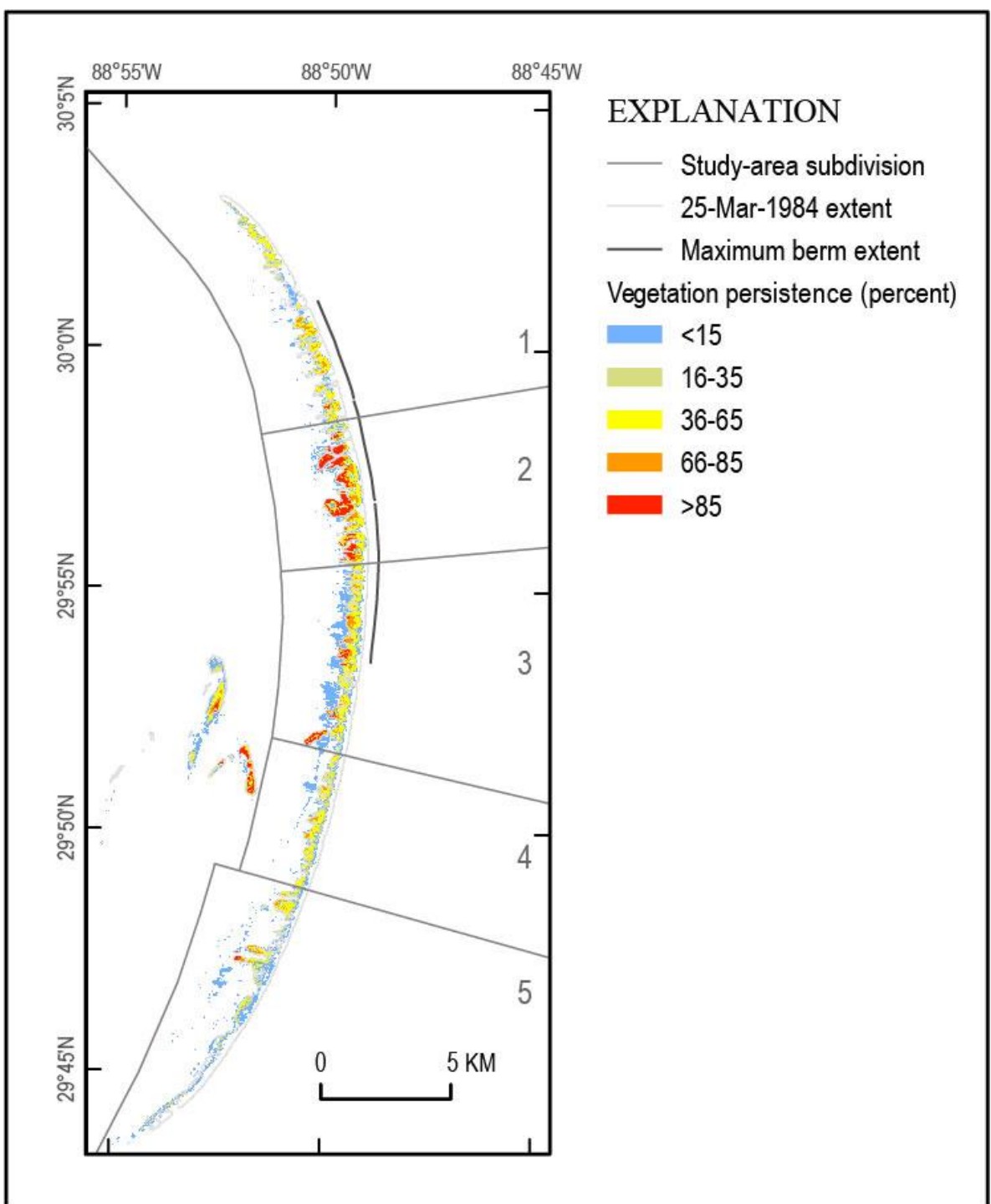

**Figure 9.** Vegetation persistence plot for the northern Chandeleur Islands represents the percent of classed land-cover images from this study for which each pixel was classed as vegetated for the entire analysis period (25-March-1984 to 19-January-2019; N = 75). Imagery is overlaid with study-area subdivisions (Table 1); 25-March-1984 barrier-platform extent is shown for reference.

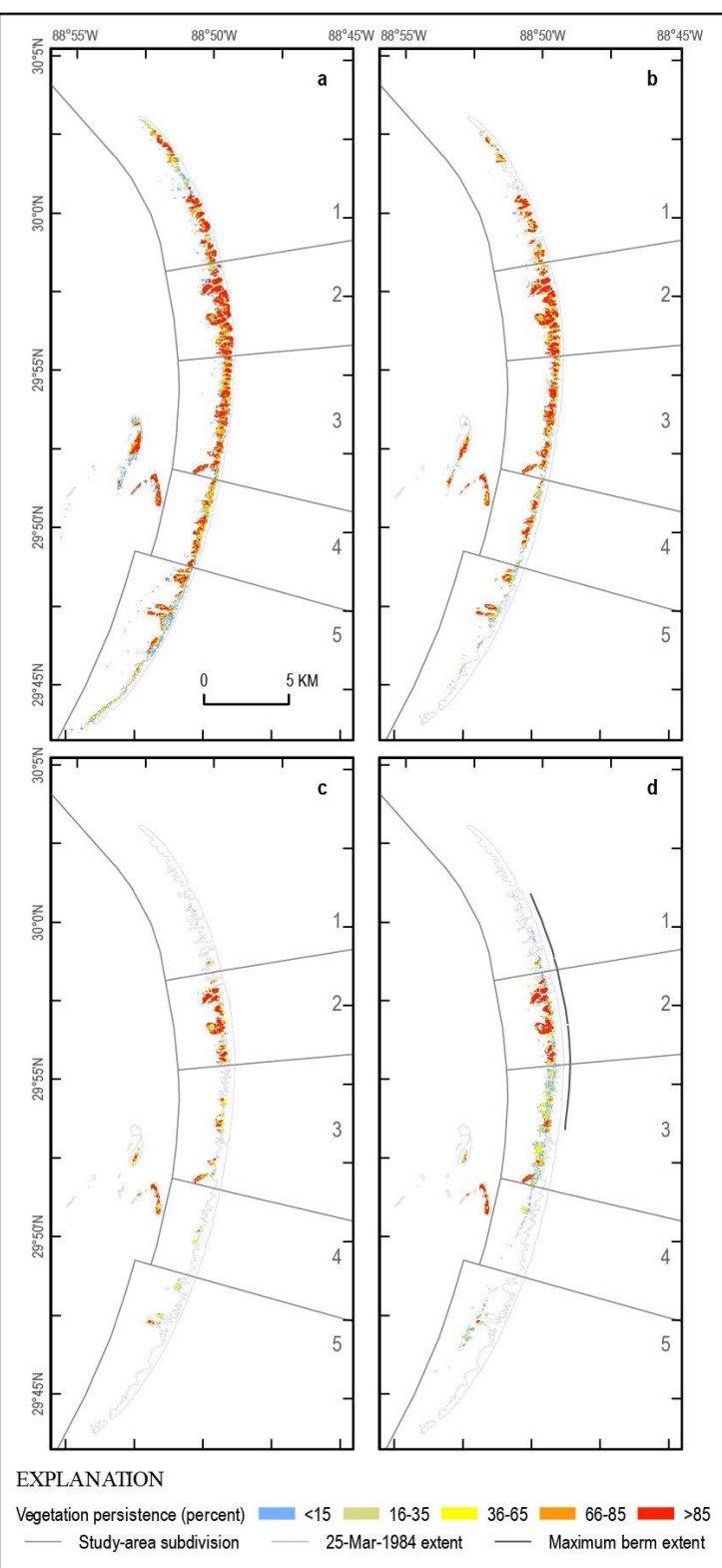

**Figure 10.** Vegetation persistence plots for the northern Chandeleur Islands represent the percent of classed land-cover images from this study for which each pixel was classed as vegetated for (**a**) 25-March-1984 to 22-April-1998 (pre-Hurricane Georges; N = 27), (**b**) 10-January-1999 to 24-March-2005 (pre-Hurricane Katrina; N = 23), (**c**) 18-October-2005 to 18-Febuary-2010 (post Hurricane Katrina; N = 8), and (**d**) 3-December-2010 to 19-January-2019 (post-berm construction; N = 17). Imagery is overlaid with study-area subdivisions (Table 1); 25-March-1984 barrier-platform extent is shown for reference.

### 3.4. Vegetation Persistence

Vegetation persistence was analyzed for the entire analysis period as well as the time periods defined in Table 3. Analysis of all 75 images acquired between 1984 and 2019 (Figure 9) demonstrates the occurrence of a persistent vegetated (marsh) platform in subarea 2 compared with the rest of the study area. Temporal changes in vegetation occurrence and persistence are shown in Figure 10. Along the length of the study area, the vegetated platform narrowed from the seaward edge, although the timing of vegetation loss varied alongshore. In subareas 1 and 5, post-Hurricane Georges vegetation losses occurred where pre-Georges vegetation persistence was lowest (Figure 10a), and the remaining vegetated platform narrowed and became fragmented (Figure 10b). Following Hurricane Katrina (Figure 10c), vegetation losses in subareas 1, 4, and 5 were extensive, the vegetated platform in subareas 2 and 3 narrowed and, in subarea 3, became highly fragmented. By 2010 (Figure 10d), except for subarea 2, very little vegetated area persisted within the 1984 barrier-platform footprint. Between 2010 and 2019, newly vegetated areas emerged in subareas 3, 4, and 5, in most cases well behind the 1984 back-barrier shoreline.

## 4. Discussion

### 4.1. Automatic Thresholding of Mulitple Spectral Indices for Rapidly Assessing Coastal Metrics

Our work demonstrates the applicability of performing automated thresholding with multiple spectral indices for delineating land-cover classes and feature extents. Compared with the use of static thresholds [37,81,82,84], dynamic histogram thresholds reduce classification uncertainties stemming from spectral variability related to changes in instrumentation and/or differences in atmospheric or seasonally-variable hydrologic and phenologic conditions between acquisition dates [95,96]. Further, the methods described here allow for rapid classification and delineation of barrier-island extents from a large number of source images relative to more time-consuming object-based [39,97], supervised [38], and unsupervised [35] classification techniques, which require training data to guide the classification and (or) "expert knowledge" to group similar clusters. Our resulting multi-decadal, high temporal resolution dataset provides a basis for better understanding the timing, nature, and potential drivers of landscape change [98] compared with assessments that quantify discrete changes between just a few points in time [38,39,91].

Overall classification accuracy comparing water, bare earth (sand), and vegetated classes for most of the datasets considered was comparable to accuracies reported for NLCD datasets ($\geq$ 70%) [50]. Table S4 and visual comparison of the reference data with source imagery and results from this study (Figures S4 and S5) identify sources of apparent classification error for four assessments with overall classification accuracies less than 60%. Figure S4b,c, for example, show that NLCD 2004 mapped a more continuous sandy (beach) extent than our methods from the same source image. In contrast, NLCD 2006 and NLCD 2008 include sandy extents that are seaward of emergent land areas visible in the source imagery (Figures S4d,e and S5b,c). This is likely related to the fact that NLCD classification algorithms were developed across broad, regional extents of the conterminous United States [50] and, therefore, may not be as accurate at the barrier-island scale [97]. We find this is particularly true when the timing of NLCD datasets coincides with the impacts of extreme storms in this highly dynamic coastal system. Compared with NLCD 2006, which mapped discontinuous sandy extents seaward of our satellite-derived seaward shoreline, the 2005 BICM data [45,91] included only minimal sandy extents along the length of the northern Chandeleur Islands (Figure S4f,g). Consequently, both the user's and producer's accuracy for the bare earth (sand) land-cover class was 0%. This may be a function of spectral differences and pixel resolution between Landsat and the source 4-band CIR ortho-imagery used by BICM (Figure S4d,f) as well as differences in classification methods.

Overall classification accuracies comparing water, sand, vegetated, and intertidal classes were about 11% lower than comparisons that excluded intertidal areas. The predominant errors driving this varied among the datasets analyzed, illustrating the complexity of mapping intertidal extents. For example, Enwright et al. [39,97] used a rule-based classifi-

cation methodology that incorporated tidal and elevation data, probability analyses, image segmentation, and photointerpretation to map intertidal extents from high-resolution aerial imagery in the northern Gulf of Mexico, whereas Wang et al. [99] applied an algorithm that evaluated Landsat-derived mNDWI concurrently with vegetation indices and utilized frequency maps to account for variations in tidal levels and phenology to map annually-averaged tidal-flat extents across coastal China. Although we tried to minimize water-level effects by using images acquired under predicted low water conditions, both actual water levels and water clarity affect the intertidal extents and subclasses mapped using the methods described in this paper. Upon visual inspection, the thresholding results compare well with the source imagery (Figure 2), with "submerged" intertidal extents appearing spectrally more similar to open water pixels and "emergent" intertidal extents appearing spectrally more like bare earth (sand) pixels. Comparison with elevation data (Figure S2) provides additional support that the classification is valid: most elevations mapped to intertidal pixels fall between mean lower low water (MLLW) and mean high water (MHW), and there is good separation between the "submerged" and "emergent" subclass elevations. We interpret that the "submerged" and "emergent" subclasses are similar to the irregularly exposed bottoms and regularly flooded bottoms [100] habitat classes mapped by Suir et al. and Suir and Sasser [37,41].

It is important to note that the methods presented here could not separate emergent wetlands from vegetated tidal flats or dune vegetation. In part, this is a function of spatial resolution; 30-m resolution products are too coarse to map barrier habitats such as dune vegetation that may be only a few pixels wide [97]. Consequently, there is a fundamental difference between the high temporal resolution, landscape-scale land-cover products described in this study and detailed, high spatial resolution habitat maps that commonly integrate multiple data sources (e.g., field, tidal, and elevation data as well as high-resolution imagery) but are updated only infrequently [33,34,39,97]. Our results suggest, however, that the use of data that accounts for temporal continuity such as vegetation persistence maps (Figures 9 and 10) or frequency maps [99] can be used to differentiate between ephemerally vegetated tidal flats and persistent marsh platform and could help refine definition and classification of intertidal extents.

Finally, the positional accuracy of the vector shoreline, sand, and vegetated extents generated by thresholding and contouring spectral indices was not evaluated in this study. There are no contemporaneous field measurements of shoreline position from the northern Chandeleur Islands against which the satellite-derived sea shorelines could be compared, unlike analyses presented by Hagenaars et al., Pardo-Pascual et al., Nelson and Miselis, or Vos et al. [54,56–58]. Positional accuracies reported in these studies for medium-resolution Landsat and Sentinel satellite-derived shorelines along sandy coastlines are less than about one-half to one pixel seaward of the GPS shoreline, which is consistent with visual comparison of Landsat-derived sea shorelines from this study with reference datasets and available high-resolution aerial imagery. Although there is an extensive database of lidar-derived sea-shoreline data from the study area [18], direct comparison cannot be made between these data, which represent MHW shoreline positions, and the shorelines presented here, which were derived from Landsat source imagery acquired at or near low tide.

*4.2. Implications for Barrier-Island Evolution and Resiliency*

This work quantifies changes to the whole barrier system, including sandy (beach) and back-barrier (marsh and tidal flat) components, at annual to decadal scales, addressing knowledge gaps in decadal-scale barrier changes that result from incomplete historical records as well as the historical bias for focusing on shoreline and beach components without explicitly including the connectivity among barrier environments [1,101]. In doing so, we provide metrics that help better capture the timing and controls on the morphologic evolution of the northern Chandeleur Islands. The number of images and the span of time analyzed (75 images analyzed over 35 years) represent an improvement over recent

studies that have also measured landscape-scale changes to barrier systems [33–39]. This increased resolution has allowed for more robust estimates of landscape-change rates, insight into deviations from past storm and recovery cycles, assessment of the contribution of anthropogenic addition of sediment, and improved understanding of the timing and extent of transitions between barrier island morphologic states.

The spatial and temporal scope of this analysis allowed us to put storm response and recovery at the northern Chandeleur Islands in historical context. Consistent with earlier studies [15,27,37], we measure significant sea-shoreline retreat and substantial decreases in land-cover extents and total island area following Hurricanes Georges and Katrina. Fearnley et al. [15] attributed accelerated land loss and shoreline erosion at the Chandeleur Islands between 1996 and 2005 to increased tropical storm activity, culminating with extreme shoreline retreat and reductions in land area after Hurricanes Ivan (2004) and Katrina (2005) impacted the islands. The expanded temporal and spatial coverage of our data provides a new perspective. First, changes to total island area described in this study do not show clear linear land-loss trends similar to those described by [15]. Instead, island-area changes show somewhat cyclical trends during the study period that can be related to storm–recovery cycles in which sand is removed from beach and shoreface environments during storms and then gradually returns during quiescent periods. This cyclicity can lead to overestimates of land-area changes when only a portion of the cycle is considered. For example, based on linear regression of 5 island-area measurements between 1996 and 2005, Fearnley et al., (2009) [15] determined a land-loss rate of $-1.01$ km$^2$/y (r$^2$ = 0.77). Using all of our satellite-based data during the same period (29 images) results in a total island-area (sand plus vegetated extents) loss rate of $-0.69$ km$^2$/y (r$^2$ = 0.52), highlighting the importance of increased temporal resolution for calculating accurate short-term land-area change rates. Further, if we expand the regression to include the entire study period, effectively ignoring shorter-term storm-recovery cycles, we find the long-term, multi-decadal total-island loss rate is $-0.39$ km$^2$/y (r$^2$ = 0.65). These differences suggest that land-area time series that are biased toward storm impacts may significantly overpredict land-loss rates and the timing of barrier-island morphologic state changes.

Simultaneous assessment of marsh persistence and shoreline change underscores the importance of back-barrier vegetation as a control on barrier island morphologic evolution [30,31,102] and, more specifically, storm-recovery cycles. For example, our analysis shows that changes to total island and barrier-platform areas closely follow changes in vegetated extents (Figure 3b). Additionally, where the barrier island is backed by a persistent marsh platform, the seaward shoreline underwent the least amount of overall shoreline retreat during the study period and the location of the back-barrier platform edge was relatively stable where controlled by the back-barrier vegetation extent (Figure S6b). After Hurricanes Georges and Katrina, vegetation losses mostly occurred where the vegetated extent was narrow and pre-storm emergent vegetation was less persistent, resulting in conversion of emergent vegetation to unvegetated intertidal areas (Figure S6c). Narrowing of vegetated extents also occurred through erosion of the seaward marsh platform (Figure S6d). The spatial distribution of vegetation loss and persistence are reflected in alongshore shoreline-change trends: following Hurricanes Georges and Katrina, the highest shoreline-erosion rates occurred where the vegetated platform incurred the greatest losses. Conversely, net shoreline accretion was observed along transects backed by or adjacent to persistent, remnant marsh platforms, consistent with accretion of sediment of upper shoreface and beach sediment through welding of nearshore sand bars to the marsh-backed barrier platform [27,102]. Our results are consistent with recent modeling studies that emphasize the coupling of sandy and marsh environments in barrier evolution [30,103] and suggest that restoration efforts that consider this connectivity in the restoration design may be more likely to achieve increased barrier-island resiliency.

The placement of approximately 3.1 million m$^3$ of sediment [72] between June 2010 and March 2011 during berm construction was a notable departure from historical vegetative controls on barrier-system morphologic behavior. Since berm sediment was not placed

for barrier restoration, we cannot evaluate the "success" of placement, but we can identify changes that likely resulted from the addition of sediment. After construction, reworking of berm sediment by both fair-weather and storm-driven wind and waves resulted in significant alongshore and cross-shore sediment transport [37,41,59,60,63,74]. Although our dataset includes a two-year gap between useable images acquired 3-October-2011 and 24-October-2013, visual assessment of SLC-off gapped Landsat 7 and SPOT satellite images indicate that by August-2012, no subaerial expression of the northern berm remained, consistent with observations by Plant and Guy [104,105] and Suir and Sasser [37]. Similar to Suir et al. and Suir and Sasser [37,41], we observed post-2010 increases in intertidal and emergent land areas, including vegetative cover, adjacent to and north of berm emplacement. Based on increases in sandy extents we observed in subarea 1, where the berm was constructed on the submerged barrier platform, and observations reported by Miselis et al. (2021) [62], it seems unlikely that the magnitude of subaerial re-emergence observed would have been possible without the addition of berm sediment to the system. The rapid reconfiguration and degradation of the original berm (Figure S7b,c) is likely responsible for increases in shoreline-erosion rates in subarea 2 after 2010; since 2013, the sea-shoreline position along this extent has been relatively stable (Figure S7d,e). Comparison of 2010–2019 shoreline positions and shoreline-change rates show increased barrier-island stability where berm sediment was placed on the emergent island relative to locations south of berm construction. In total, berm emplacement seems to have buffered sea-shoreline position in the central part of the study area and contributed to increases in barrier-platform area in the northern part of the study area. Much of this, however, consists of unvegetated tidal flats and sandy extents that are readily reconfigured during storms, making temporary any sub-aerial ecosystem services it might provide.

Additionally, analysis of island responses south of berm placement suggests that natural post-storm barrier recovery was occurring simultaneously with berm-related changes described above, suggesting that not all post-2010 changes can be attributed to sediment placement alone. Compared with the previous analysis period, we observed post-2010 increases in intertidal and sandy extents as well as newly vegetated areas south of berm emplacement where little to no deposition of berm sediment was expected due to the dominant northward direction of LST along the berm extent [68,69]. These trends suggest that berm sediment enhanced natural recovery processes that were already taking place, making it difficult to quantify the contributions of berm sediment alone.

The narrowing of vegetated environments after Hurricane Katrina compared to previous time periods (1984–1998 and 1999–2005) and the weaker dependence of total barrier-platform width on vegetated extent since 2010 suggests that future storm-related land losses may be more extensive than in the past and could further threaten what vegetated cover still exists. Since increases in vegetated extent are lagged with respect to sandy extents, increases in storm frequency and intensity that might occur with climate change [106–108] could further reduce existing vegetated extents and deter revegetation processes, reducing island stability and prolonging recovery cycles. Also, the increased dominance of intertidal areas and the modest recovery observed since Hurricane Katrina indicates that the Chandeleur Islands might be in transition to a new morphologic state in which more of the available sediment is submerged. This could be interpreted as a transition to a submerged shoal, as suggested by other authors [15,44,102], but the temporal and spatial scope of our dataset captures response modes with more complexity. The region south of the LST node has degraded and reformed several times, each time occupying a footprint more landward of the one before it (Figure S8b). Importantly, despite narrowing and back-stepping over the study period, the newly emergent areas do become vegetated, suggesting that overwash fluxes and vegetation-aggradation rates may be in balance or very close [30,109]. In contrast, the northern-most portion of the study area was submerged and, assisted by the addition of berm sediment, has reemerged within the historical barrier-platform footprint (Figure S8c) but has remained largely unvegetated. The repeated reemergence of the island within the 1984 footprint suggests that accommodation (the space available

for sediment accumulation) is sufficiently filled to promote sub-aerial emergence. This could be related to shoreface sediment supplies, which are relatively high in comparison to the southern part of the study area [70,110], or the recent anthropogenic addition of sediment. Regardless, sediment supplies are too low to stabilize the barrier platform enough to promote vegetation growth. Finally, the positional stability of the central portion of the study area (Figure S8c), which is backed by a stable marsh platform, is consistent with modeled long-term barrier behavior (e.g., $10^2$–$10^3$ y), in which back-barrier marshes slow down the rate of landward barrier-island migration by filling up accommodation in the back-barrier lagoon [30,109]. These differences in historical response have important implications for future barrier island response to RSLR, particularly since so many regions of the study area seem to be very close to crossing morphologic thresholds. Assuming no other human alterations to the barrier system, we expect that rapid increases in RSLR will increase the rate at which behaviors observed in the more dynamic portions of the barrier island cascade to areas of historical stability [111] and that observed spatial variability in vegetation persistence and sediment supply will continue to control the nature and timing of barrier island morphologic state transitions in the future.

## 5. Conclusions

The results presented in this study demonstrate that automated thresholding algorithms can be applied to multiple spectral indices derived from medium-resolution Landsat satellite imagery to rapidly delineate land-cover classes and barrier-island extents at the landscape scale. Analysis of this high temporal-resolution dataset provides better understanding of barrier response to past perturbations, revises previous interpretations of the response at the study site, and provides insight to the potential future evolution of the northern Chandeleur Islands. Our results reveal alongshore-variable patterns of landscape response to both natural (storm) and anthropogenic (berm emplacement) perturbations at annual to decadal scales and provide new data that demonstrate the importance of vegetative controls on barrier shoreline change, transgression, and landscape evolution:

- Land-cover changes show decadal-scale oscillations related to storm–recovery cycles. Linear trends derived from shorter and (or) less resolved time series are biased toward storm impacts and may significantly overpredict land-loss rates and the timing of barrier morphologic state changes.
- Patterns of landscape change and recovery following storms varied alongshore and were directly related to vegetation extent and persistence trends. Although redistribution of emplaced berm sediment contributed to post-2010 increases in intertidal and emergent land areas, natural post-storm barrier recovery was occurring simultaneously with berm-related changes, making it difficult to decouple the natural and anthropogenic contributions.
- We show a transition from a persistent, emergent vegetated (marsh) platform to newly vegetated back-barrier flats landward of the historic barrier footprint and interpret that this implies increased vulnerability of the extant landscape to future storms and RSLR.

**Supplementary Materials:** The following are available online at http://www.mdpi.com/s1. Table S1: Tropical systems that passed within 200 km of the study area between 1979 and 2019; Table S2: Spectral bands for Landsat 5 Thematic Mapper (TM), 7 Enhanced Thematic Mapper Plus (ETM+), and 8 Operational Land Imaging (OLI) sensors used in this study; Table S3: Source image-acquisition dates used to create reference datasets used in accuracy assessment; Tables S4 and S5: Confusion matrices showing results of accuracy assessment; Table S6. Land-cover areas for the entire northern Chandeleur Islands study area averaged for time periods corresponding to natural (storm) and anthropogenic (berm construction) events that affected historical land-cover changes; Figure S1: Study-area map showing cross-shore transects used to calculate barrier metrics and shoreline-change rates; Figure S2: Boxplot showing distribution of elevations extracted from temporally consistent lidar datasets for intertidal "submerged" and intertidal "emergent" subclasses; Figure S3: Linear-regression shoreline-change rates along cross-shore transects spaced 300 m alongshore; Figures S4 and S5: Maps comparing reference data and source imagery with classed results from this study;

Figures S6–S8: Maps showing vector shoreline positions and feature extents demonstrating changes that occurred during the study period.

**Author Contributions:** Conceptualization, J.C.B.; methodology and validation, J.C.B.; formal analysis, J.C.B. and J.L.M.; data curation, J.C.B.; writing—original draft preparation, J.C.B.; writing—review and editing, J.C.B., J.L.M., and N.G.P.; visualization, J.C.B.; supervision, J.L.M.; project administration and funding acquisition, J.L.M. and N.G.P. All authors have read and agreed to the published version of the manuscript.

**Funding:** This research was funded by the U.S. Geological Survey Coastal and Marine Hazards and Resources Program.

**Data Availability Statement:** Data generated during this study are available as a U.S. Geological Survey data release (Bernier, 2021, "Coastal Land-Cover and Feature Datasets Extracted from Landsat Satellite Imagery, Northern Chandeleur Islands, Louisiana, https://doi.org/10.5066/P9HY3 HOR") [93].

**Acknowledgments:** The authors thank Tim Nelson for reviewing the methodology and data products that that accompany this manuscript. Scientific reviews by Dan Ciarletta, Kathryn Weber, and 2 anonymous reviewers provided constructive feedback that improved the manuscript. Any use of trade, firm, or product names is for descriptive purposes only and does not imply endorsement by the U.S. Government.

**Conflicts of Interest:** The authors declare no conflict of interest. The funding agency had no role in the design of the study, in the analyses or interpretation of data, in the writing of the manuscript; or in the decision to publish the results.

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
