# Peer review of "Satellite-Derived Barrier Response and Recovery Following Natural and Anthropogenic Perturbations, Northern Chandeleur Islands, Louisiana"

_remotesensing, doi:10.3390/rs13183779_

Round 1
Reviewer 1 Report
The authors explore the relative contributions of storms and human alterations to sediment supply on decadal changes in barrier landscapes using multiple satellite-derived spectral indices for coastal land-cover classification and analyzed Landsat satellite imagery to quantify changes to the northern Chandeleur Islands barrier system since 1984. The results are accurate and interesting to the reviewer. Some questions and comments are
- Automatic Otsu’s thresholding is applied to segment satellite images into land and water areas and delineate shorelines. The threshold corresponds to the histogram of different bands to classify two different objectives. In the second step, are the determined thresholds in the NDVI index remarkably different from or close to those in the NBVL index for the whole images? Does the marked sand area in the NBVL image overlap with the vegetated area in the NDVI image?
- Do the incident waves break to induce whitecaps near the shoreline that possibly be mistaken as sand instead of water? If yes, how to treat with the whitecaps?
- Images were collected within 2 hours of predicted low tide or were collected on a rising tide with predicted water levels at time of acquisition less than MSL in this study. The tidal range is small with a mean of 0.42 m. If the beach slope is small, tidal variation has significant effect on determining shoreline change from different images. How about the beach slope?
Reviewer 2 Report
This paper used a thresholding method to automatically classify remotely-sensed images of the Chandeleur Islands that were collected using multiple platforms between 1984 and 2019. They also used AMBUR to evaluate changes in the location of the shoreline and other borders over time. The overall approach appears to be sound, with classification accuracies of the main three classes (vegetation, water and bare earth) ranging from 50 to 80%. The analysis is comprehensive and shows changes in land-cover extent and the location of various borders over a time period that covers two major hurricanes (Georges and Katrina) and construction of a berm. This is all fine as far as it goes, and the manuscript provides a detailed description of this area. The introduction and methods are well-written. However, I had a lot of trouble with the nomenclature and I suggest the authors consider dropping the submerged intertidal classification since it performed so poorly. I would also like to see more quantitative analyses.
Here are my major questions/suggestions:
- The classification “vegetation” appears to be restricted to upland vegetation only, given that these areas were masked before the last classification into intertidal (submerged vs. emergent). Is this correct? If so that needs to be explicitly stated. Or does “vegetation” sometimes include emergent intertidal vegetation? (I was confused because I was looking for information about the extent of the vegetated marsh and this was not called out.)
- The terminology was confusing. What is meant by “non-water land area”? This is an awkward term and I’m not sure what it includes. Does this mean total land area (e.g. the sum of bare earth and vegetation)? Does it include the emergent intertidal areas? Whatever it is needs to be defined and perhaps there is a better term. And how is this different from “Island” area, “total land”, and “barrier platform”? Is sand different from mud? (that is, is it meant to be on the beach side only?). Which categories are included in “subaerial”? Also, if “back-barrier platform” includes both marsh and tidal flat, can these be separated out? Can the beach be separated from the mud flat on the back side? How do remnant marsh platforms come into play/intersect with this? I’m sorry if I sound cranky – it was just really hard to follow what was being done with so many different terms being used.
- The classification of intertidal areas is challenging and I’m not convinced that it worked. It is understandable that submerged areas would be confused with water and that emergent areas are tricky to define. However, this issue is not acknowledged until the Discussion and the classification accuracies for these categories are buried in the Supplement. I had notes all over the manuscript wondering about the intertidal areas until I got to this part. Please address this in section 3.1. This also has implications for the discussion of the back-barrier characteristics, which is touted as one of the benefits of this analysis. If the classification accuracies are low this needs to be acknowledged as a limitation of this method – and perhaps the submerged intertidal classification needs to be dropped since those accuracies were so low and there were so few ground-truth observations. This might also affect the confidence placed in the analyses of Figures 9 & 10 (lines 390-403).
- This is a rich data set and the introduction provides several examples of how it might be used. However, what we have is a very detailed description and very little quantitative analysis (beyond Figure 8). Can the results of this effort be used to look at connectivity? Or to provide insight into the relationship between marsh persistence and shoreline change? (line 529). Line 499 talk about metrics regarding morphological evolution but I’m not sure what they are. Are there landscape change rates that can be looked at (beyond the AMBUR shoreline?)
- The set-up introduces a suite of potential drivers of barrier systems, not all of which are dealt with (e.g. sediment supply, RSLR). There is a discussion of the effect of the berm sediment, but in the end the contribution of the sediment addition cannot actually be separated out. I would suggest backing off of the first few lines of the abstract (lines 10-14) and introduction (lines 35-38) unless the analysis yields insights that can be tied back to these ideas.
- Along those same lines, there is a mention of “significant decreases” in lines 275 and 276. Were these statistical tests? If not I’d avoid the language – but better yet to actually test for significant changes here and throughout where it makes sense.
- I’m not sure I understand why having higher temporal resolution yields such a different answer in terms of land-area change rates (lines 517-528). Adding data points should tighten up the errors (e.g. reduce variability of the estimate), but not necessarily bias it. It is certainly fair to say that basing a rate on points that are all relatively high would produce a high number, but one could argue that a few observations that were all relatively low would produce a low number. In this case if the point is that the observations did not span the same period and that if you had done the estimate based on just the initial response after the hurricane (e.g. a shorter time period) you get a different answer, then that is a different issue. Please make sure that you are comparing apples to apples.
Line comments:
This may just be the journal’s style, but it is really frustrating to refer to a paper by number when it is part of the sentence as it just forces you to either look it up or skip/miss the point. Examples of this are line 180 [85], line 185 [87], line 515 [15], line 520 [15], line 562 [62]- as compared to Line 187 which does say “Estoque and Murayama [90]” or line 510.
There are 110 references. That is a lot and would be outside the limit for many journals.
Line 23: last couple “of” decades
Line 89: new insight “into” alongshore…
Line 89: I think you mean “complement”
Fig. 1: Is the black box in (a) the area that is blown up in (b) and (c)?
All Figures: Why do you say “Explanation”? I’ve never seen that before.
Fig. 2: Here is where I started to wonder if “Vegetated” should be “Upland vegetation”, as opposed to “intertidal”, and also whether either of the intertidal categories included vegetation.
Line 236: Again, did this analysis include intertidal areas?
Line 292: “Infrequent, land area stabilized” is awkward
Fig. 3. “Intertidal” is misspelled in the legend here and in Figs. 4 and 5. “Total Isand” is perhaps meant to be “Total Island?” although I’m not sure how this is different from “Total Land”. Also, how is Panel c different from Panel a? Is one normalized differently or is c just a close-up? (not clear in the caption)
Table 4. How was this calculated? Were there linear regressions performed for each transect within a section that were then averaged? What are the errors on these averages? (Also, Figure S2 is quite interesting – I might suggest including it in the main paper.) In addition, please make it clear which side of the Islands you are on for this estimate (landward or seaward?).
Figure 6 – Figure 6 is described in the text (lines 325-326) as showing the positions of the edges of the back-barrier island, the demarcation between the sand and the vegetation, and that between the sea and the shoreline. 1) Can each of these be more clearly defined? Is Figure 6a the edge between the water and the beginning of the submerged (or emergent?) intertidal area? Or between the intertidal and the upland vegetation? Is Figure 6b, the “sand-vegetation” line, on the open ocean side? And is the “sea-shoreline” in Figure 6c the border between the beach sand and the ocean? Note that Figure 6a is labelled (incorrectly?) as “back-barrier extent”, and that neither Figure 6a nor 6b is referred to in the text. It seems like it would be useful to use this as an opportunity to show the locations of a) the inshore/intertidal border, b) the intertidal/upland vegetation border, c) the upland vegetation/sand border, and d) the sand/offshore border. Even better to separate out the location of submerged vs. emergent intertidal if you have confidence in those categories.
Figure 7 – I did not find this figure helpful. It is almost impossible to connect bars of a particular color, and the idea that no bar might mean that either that feature was not present (e.g. width = 0), or that there were not enough observations, makes the lack of a bar uninformative. I would remove this figure, and consider a table with average widths of each feature within each section. I again need clarification as to what is considered the “barrier platform”, whether “beach” is equal to “sand” and means it is on the offshore border, whether “vegetation” means upland, and whether it would make sense to add a category for “intertidal” or “marsh” extent. (Again, I am really confused and perhaps “back-barrier platform extent” is meant to represent the marsh?)
Figure 8 – Did you consider doing these regressions against back-barrier platform extent, and/or against the sum of vegetated upland and emergent intertidal? This might help address the speculation in line 379 about increased dominance of intertidal areas.
Lines 396-400: This is an example of qualitative observations that could potentially be quantified (e.g. comparing vegetation persistence, platform narrowing and fragmentation in each section).
Line 402 – Does newly vegetated areas mean upland? Just thinking about colonization of intertidal areas.
Figure 9 – Surely the legend can be condensed, with % and number of occurrences both represented next to a color panel. This makes it look like there are 10 different categories. Also, this discusses a persistent vegetated marsh platform. Is it based on areas classified as intertidal emergent?
Line 421: “Phenologic”
Lines 447 – 470: All of this belongs in the results.
Line 450: substitute “among” for “between”
Lines 480-481: This brings up an interesting idea (assuming the classification ends up being robust for intertidal areas), which could be added to the analysis as a way to demonstrate its utility.
Lines 482-493: This also seems to belong in the results
Line 492: “and the shorelines presented here, which were derived from”
Line 514: “provides a new perspective”
Line 538: here is an example where it might be possible to quantify either the area or rate of conversion from emergent vegetation (= emergent intertidal?) to barren area (= sand?). Also, it is not particularly surprising to see that shoreline erosion occurred where vegetated platform was lost – that is essentially the definition of what erosion means (Lines 541-542).
Lines 543-547: Is there any way to support or test these ideas quantitatively?
Line 576: Here is another example of where I didn’t know what to evaluate in terms of intertidal and sandy extents. Does intertidal mean the sum of emergent and submerged? Is sand just the beach?
Lines 599, 604-5: “accommodation is sufficiently filled’ and “filling up accommodation” must be a term of art that I am not familiar with. Can you please clarify what you mean by accommodation?
Lines 616, 618: For the Conclusions (which is all some people might read), it seems better to say barrier “island”
Lines 623-625: Again, I can understand why shorter time series might yield a higher estimate if it is focused on a storm event, but I don’t think the idea of “less resolved = higher estimate” is supported.
Line 886: I did not check all of the references, but I did notice a typo in this one, “Systmes”
